# Polyurethane-Encapsulated Biomass Films Based on MXene@Loofah Sponge for Piezoresistive Pressure Sensor Applications

**DOI:** 10.3390/polym16101377

**Published:** 2024-05-12

**Authors:** Qihan Jia, Shuai Liu, Haibo Wang

**Affiliations:** College of Biomass Science and Engineering, Sichuan University, Chengdu 610065, China; jiaqihan@stu.scu.edu.cn (Q.J.); 2021323080023@stu.scu.edu.cn (S.L.)

**Keywords:** MXene, silver nanowires, loofah, pressure sensor, synergistic conductive network

## Abstract

Multifunctional wearable electronic sensors exhibit significant potential for applications in health management, motion tracking, intelligent healthcare, etc. In this study, we developed a novel assembly method for a polymeric silver nanowire (Ag NW)/transition metal carbide/nitride (MXene) @Loofah device using a facile solution dip-coating technique. During the pretreatment phase, the loofah was conditioned with polydiallyldimethylammonium chloride (PDAC), promoting the self-assembly of MXene layers and bolstering device stability. Then, the Ag NWs/MXene@Loofah was packaged with polyurethane to form a piezoresistive pressure sensor, which demonstrated superior pressure-sensing capabilities and was adept at registering movements of human joints and even subtle pulses. The design strategy presents a novel and rational approach to developing efficient pressure sensors.

## 1. Introduction

In recent years, significant strides have been made in developing wearable electronics [1,2,3,4,5,6]. Their portability and shape versatility endow them with outstanding performance and hold the promise of further advancements across various fields. As societal activities become increasingly complex, there is a growing demand for wearable electronic devices that seamlessly integrate high performance with versatility [7,8,9]. Therefore, it is of paramount significance to discover a portable wearable electronic device characterized by a stable interface, diverse functionalities, and a simple structure that utilizes multipurpose materials. Studies have revealed that flexible sensors based on a singular morphological material often struggle to achieve a balance between high sensitivity and a wide induction range due to inherent structural limitations [10,11]. For instance, zero-dimensional materials such as metal particles are prone to separating from each other, thereby leading to a broad induction range. Conversely, sensors fabricated from one-dimensional materials, such as silver nanowires (Ag NWs) and carbon nanotubes (CNTs), typically exhibit a wide response range but suffer from low sensitivity [12]. This limitation predominantly arises from their high aspect ratio, which impedes noticeable resistance changes even under substantial deformation. In the case of two-dimensional (2D) materials, the close stacking and interaction between adjacent layers impede the generation of effective slip when subjected to force, often leading to large cracks that disrupt the conductive pathway [13]. Consequently, sensors typically exhibit high sensitivity within a very narrow induction range. In the present study, we focus on 2D nanomaterials, particularly transition metal carbides/nitrides (MXenes), which are renowned for their layered structure, excellent conductive performance, high specific surface area, and robust processability [14,15]. MXenes have found wide-ranging applications in electrochemical energy storage, sensor technology, and electromagnetic interference shielding [16]. Incorporating MXenes into sensors has the potential to enhance sensing sensitivity and broaden the scope of sensing capabilities [17]. Additionally, the liquid-phase etching process produces rich polar functional groups (-OH, -O, and -F), rendering MXenes hydrophilic and facilitating the formation of uniform and stable water suspensions [18]. Its compatibility with solution-based methods, such as spraying and soaking, makes it an ideal candidate for developing highly sensitive strain sensors. However, weak adhesion between MXene sheets predisposes them to sliding and crack expansion during cyclic compression. This results in the rapid degradation of the conductive network of sensors and a subsequent decline in performance [19].

Moreover, the electrical conductivity of a single MXene coating is not ideal, and its structure cannot inherently exhibit both high sensitivity and a wide induction range. It is commonly remedied to address this limitation by introducing a second phase to the MXene material or by regulating the morphology of MXene [20,21,22]. However, due to the random size and agglomeration defects of MXene, assembling MXene nanosheets into macroscopically interconnected conductive networks proves challenging [23]. Ag NWs possess inherent metal conductivity and an interwoven “fishing net”-like conductive structure, facilitating contact and the construction of a highly conductive macroscopic network [24]. Nevertheless, NW-based conductive networks exhibit poor mechanical properties and tend to slide when subjected to strain during practical applications [25]. We propose integrating Ag NWs into MXene nanosheets to overcome these shortcomings and establish a three-dimensional (3D) interconnected conductive network [26,27,28,29]. Utilizing MXene nanosheets as a scaffold strengthens the Ag NW network, while embedding Ag NWs between MXene nanosheets prevents stacking and facilitates interconnection, thereby enhancing electron transfer between nanosheets [30]. This integration ultimately improves the electrical conductivity and mechanical performance of the sensor.

In addition, due to the inherent hydrophilicity of MXenes, direct drop coating or spin coating often results in uneven thickness of the conductive film when loading MXenes onto hydrophobic substrates. A self-assembly approach has been employed to address this issue, combining MXene with polydiallyldimethylammonium chloride (PDAC) to form a composite film that can be uniformly applied onto the substrate [31]. MXene exhibits poor antioxidant performance, making it susceptible to oxidation and consequent loss of conductive properties during practical use. To mitigate this, a coating method is employed to shield MXene from the external environment. Polyurethane (PU) is widely used as a sensor package and as a substrate because of its outstanding performance advantages, such as its elasticity, flexibility, and durability. It can be hard, soft, protective, elastic, or adhesive by design and manufacture [32,33]. PU enhances adhesion between MXenes and the substrate, enabling the sensor to maintain excellent electrical conductivity over an extended period.

Sensors based on PU films are commonly used as strain sensors because of their excellent flexibility. However, the 2D structure of the films limits their application in piezoresistive pressure sensors. Constructing a 3D conductive network that can output sensitive electrical signals under pressure statements is necessary to improve the sensitivity based on the piezoresistive effect. Loofah sponges, which are renewable biomass materials, have a natural 3D network structure. The 3D network structure not only provides the support structure for the conductive fillers but also enhances the strength of the sensors.

The cost-effective and lightweight loofah sponge possesses a highly macroporous structure with an open pore network [34,35,36]. The sizable pores within the loofah skeleton and the microchannels within its fibers contribute to its high surface area. This characteristic enables effective coverage of MXene on the surface, ensuring excellent conductive performance of the sensor [37,38,39]. Moreover, the internal channels of the loofah provide resilience against breakage during rubbing and stretching. Even in instances of partial breakage under strong tension, alternate pathways exist for signal current, ensuring prolonged sensor usability [40]. This study presents the assembly of a wearable porous loofah sensor coated with PU, PDAC, MXene, and Ag NWs. Incorporating MXene and Ag NWs facilitates the formation of an efficient and stable conductive network, thereby expanding the sensing range and enhancing the mechanical properties of the sensor. Incorporating PU effectively mitigates surface oxidation of MXene and enhances device durability. Following research, the assembled sensor demonstrates outstanding electrical conductivity and is capable of translating various human movements into distinct electrical signals. These movements encompass both subtle actions such as joint bending and more pronounced expressions such as facial movements, highlighting the applicability of the sensor in wearable devices. Importantly, the sensor structure comprising MXene/Ag NWs/PU/PDAC proves both feasible and efficient, streamlining the preparation process and facilitating widespread practical application.

## 2. Experimental Section

### 2.1. Materials

Titanium aluminum carbide (Ti_3_AlC_2_, 300 mesh, >98.0%) was purchased from Foshan Xinxi Technology Co., Ltd, Foshan, China. Silver nanowires dispersed with ethanol purchased from Jining Leadernano Tech. Co., Ltd., Jining, China. The Ag nanowires had a diameter of 40 nm and a length greater than 2 μm. Lithium fluoride (LiF, 99.99%), hydrochloric acid (HCl, 12 mol/L), and anhydrous ethanol, as well as Poly (diallyl dimethyl ammonium chloride) (PDAC, 99.99%) solution, were purchased from Shanghai Titan Scientific Co., Ltd., Shanghai, China. Polyurethane (PU) (including a cross-linking agent) was obtained from Shenzhen Jipeng Co., Ltd. (Shenzhen, China), and the model was MR-913 (40% solids). Commercial cotton fabric was obtained from Dragon Clan Co., Ltd. (Guiyang, China). Conductive silver paste was obtained from Xunying Co., Ltd, Shanghai, China. Natural loofah sponges were bought from the YANCHEG OUKAI sponge product factory. Deionized water was made in the laboratory with EREERAN 1253 equipment. If not specified, the products purchased above were used directly without any other treatment.

### 2.2. Synthesis of Ti_3_C_2_T_x_ MXene

Based on proven methods reported in previous articles, the synthesis process commenced with the mixing of 3.2 g of LiF and 40 mL of HCl (9 M) in a PTFE reactor. This mixture was then heated in a heated oil bath for 5 min at 40 °C. Subsequently, 2 g of Ti_3_AlC_2_ was gradually added to the PTFE reactor, and the reaction was stirred at room temperature for 48 h to ensure complete etching of the aluminum layer. Following this, the mixture was then washed several times with deionized water via centrifugation at 8000 rpm until the pH of the system reached 5–6. The prepared multilayer MXene suspension was then subjected to sonication under an argon gas atmosphere and in an ice bath environment for approximately 2 h to facilitate further stratification. Subsequently, centrifugation at 3500 rpm for 30 min yielded a monolayer of MXene slices from the supernatant of the centrifuged product.

### 2.3. Preparation of the PAM@Loofah

The entire preparation process of the PAM@Loofah is depicted in Figure 1. First, the loofah was cleaned in an ultrasonic machine using deionized water. After 20 min, the loofah was removed and placed in an oven at 40 °C for drying. Subsequently, the process was repeated once by replacing the deionized water with ethanol. The washed loofah was immersed in PDAC solution (40 wt% aqueous solution) for 2 h and then dried at 40 °C. The process was repeated three times. After that, PDAC-treated loofahs were completely immersed in MXene suspension (10 mg/mL) for 10 min and then dried in a vacuum oven at 40 °C for 2 h. This process was repeated twice to obtain M@Loofahs. Following this, the M@Loofahs were completely immersed in an ultrasonic Ag nanowire dispersion (1 mg/mL) for 20 min and then removed and dried at 40 °C for 10 min, repeating the process twice to obtain AM@Loofahs. A silver paste with good conductivity was applied to both sides of the AM@Loofah sensor, which was then connected with copper sheets to ensure a complete conductive path. Subsequently, the AM@Loofah was immersed in polyurethane with a curing agent and cured at 40 °C for 12 h. Finally, the cured polyurethane was cut and trimmed to obtain PAM@Loofahs.

### 2.4. Characterizations

The zeta potentials of MXene dispersions and PDAC were tested using Zeta sizer NPS (Malvern Instruments, Malvern, UK). An XSAM 800 instrument was used to record X-ray photoelectron spectroscopy (XPS) spectra. The X-ray source was a 1486.6 ev Al Kα excitation source. X-ray diffraction (XRD) was recorded with a Rigaku Smart Lab using a 1.54178 Å Cu Kα excitation source. The recording range was from 9° to 90°. The morphology of the MXene, loofah, M@Loofah, AM@Loofah, and PAM@Loofah was recorded with a Quanta 250 (FEI, Hillsboro, OR, USA) scanning electron microscope (SEM). The elemental distribution and the content were analyzed with an SEM equipped with energy-dispersive X-ray (EDX) spectroscopy. The microstructure of MXene nanosheets was recorded through field-emission transmission electron microscopy (TEM, Tecnai G2 F20 STWIN, FEI, Hillsboro, OR, USA). A Zhihe mobile device (model: 01RC, LinkZill Technology, Hangzhou, China) recorded the relative resistance change signal of the sensor and sent real-time data to a smartphone via Bluetooth. 

## 3. Results and Discussion

### 3.1. Fabrication of PAM@Loofah Devices 

Figure 1 illustrates the preparation of PAM@Loofah sensors using MXene electrostatic self-assembly and Ag NW dispersions. First, a loofah was used to build the skeleton of the sensor by immersing it in PDAC. Subsequently, MXene and Ag NWs were progressively coated onto the loofah fiber skeleton to construct a conductive network. These steps were repeated three times before being prepared for a pressure sensor. MXene nanosheets exhibited a negative potential in a uniform suspension (with a zeta potential of approximately −30.2 mV), while PDAC carried a positive charge (with a zeta potential of about +18.7 mV), facilitating the electrostatic self-assembly of MXene. The conductivity of the sensor was calculated based on its resistance and the corresponding dimensional data with the following formula:$$\mathrm{conductivity}=\frac{\mathrm{GL}}{A}$$
where *G* is the conductance of the sensor, and *L* and *A* are the length and cross-sectional area of the sensor, respectively. As shown in Figure 2a, the loofah treated with PDAC had better conductivity. This indicates that after the PDAC treatment, MXene could be better loaded onto the loofah. The electrical conductivity of the loofah without the PDAC treatment was only 0.31 mS/m. After treatment with PDAC, the electrical conductivity of the loofah reached 3.52 mS/m, which was 11 times higher than that of the untreated loofah. In addition, the conductivity of the sensor was substantially improved by the combined effect of MXene and Ag NWs. After the synergistic effect of MXene and Ag NWs, the conductivity of the filipendula reached 16.70 mS/m. This interaction formed a composite film, enhancing the conductive efficiency. As shown in Figure 2b for the continuous electrical signal stability test, the sensor signal disorder without the PDAC treatment could not maintain the same relative resistance change. The sensor signal was stabilized after using PDAC to treat the loofah. This can be explained by the increased surface stability of the PDAC-treated loofahs, which reduced sliding during repeated extrusion and minimized the formation of cracks. This result ensures the stability and long-term availability of the device.

Figure 1a depicts Ti_3_C_2_T_x_ for MXene obtained through Al layer etching and multilayered MXene sonication. As shown in Figure 2d, a clear Tyndall effect was observed by irradiating the prepared MXene suspension using a laser pointer. To further characterize the prepared MXene, Figure 2e shows the X-ray diffraction (XRD) patterns of etched MXene and Ti_3_AlC_2_ (MAX). The XRD results of Ti_3_C_2_T_X_ exhibited characteristic diffraction peaks at 2θ of 9.1° (002) and 19° (004). The XRD pattern of Ti_2_AlC_3_ shows a strong diffraction peak at 38.8° (104). However, after etching, the diffraction peak of MXene at 38.8° disappeared. This result indicates that the aluminum atomic layer in Ti_2_AlC_3_ was completely removed [41]. Furthermore, the absence of the aluminum atomic layer led to an increased spacing between the Ti_3_C_2_ layers. 

As illustrated in Figure 2f–i, the elements present in the MXene and their valence states are depicted through X-ray photoelectron spectroscopy (XPS). The results revealed signals of Ti, O, F, and C elements in the MXene. As shown in Appendix A, the XPS spectra of MXene showed characteristic peaks of Ti2p, F1s, C1s, and O1s at 456 ev, 685 ev, 285 ev, and 531 ev, respectively. Further fitting in Figure 2f reveals five distinct characteristic peaks at 468.2 ev, 462.9 ev, 460.5 ev, 456.8 ev, and 455.3 ev belonging to Ti-O (2p^1/2^), Ti^2+^/Ti^3+^ (2p_1/2_), Ti-C (2p_1/2_), Ti^2+^/Ti^3+^ (2p_3/2_), and Ti-C (2p_3/2_). The two fitted peaks of F1s in Figure 2g belong to TiO_2_-x-F_x_ (685.6 eV) and C-Ti-F_x_ (684.5 eV), respectively. In contrast, four fitted peaks of C1s appear in Figure 2h, belonging to C=O (288.7 eV), CH_x_/CO (285.5 eV), C-C (284.7 eV), and C-Ti-T_x_ (281.6 eV). Alternatively, the fitted peaks of O1s in the range of 528 to 536 eV in Figure 2i belong to C=O, C-O, and Ti-O bonds. These data provide strong evidence for the successful preparation of MXene nanosheets.

The morphology and microstructure of the sample surface were observed using scanning electron microscopy (SEM), as depicted in Figure 3a–f. Figure 3a–c display the surface structure of the loofah at various stages at low magnifications, with the enlarged views arranged in Figure 3d–f. Figure 3a illustrates the relatively rough surface of the initial unprocessed loofah fiber. Following repeated alternating soaking of the loofah fiber in the MXene water solution and PDAC, the fiber surface exhibited smoother features. Subsequently, smaller bulges were observed on the fiber surface in Figure 3c,f, representing coordinated conductive networks formed by combining Ag NWs and MXene. In addition, the SEM image of the PAM@Loofah cross-section in Appendix A shows that the conductive network is also formed in the interior of the loofah through the slits. Appendix A shows the EDS image of the PAM@Loofah demonstrating the content of each element in it. Appendix A shows the specific content of each element in the PAM@Loofah, where Ti and Ag reached 16 wt% and 15.56 wt%, respectively. This result indicates that MXene and Ag were successfully encapsulated onto the loofah. Further, it can be seen in Figure 3h–l by the distribution of the EDS mapping of C, Ti, Ag, F, and O that Ag and Ti are evenly distributed on the loofah. These results demonstrate that MXene and Ag NWs are evenly encapsulated on the loofah.

### 3.2. PAM@Loofah-Based Sensors and Performance

Good electrical conductivity is an indispensable property for sensors. As shown in Figure 4a, when the PAM@Loofah sensor was connected in series with a power supply and a light bulb, the PAM@Loofah could light the light bulb. This indicates that the PAM@Loofah had good electrical conductivity. In addition, the bulb became brighter when pressure was applied to the PAM@Loofah sensor. This indicates that the sensor resistance became smaller, and the PAM@Loofah could sense. The relative resistivity change rate (ΔR/R_0_) of the PAM@Loofah was tested in the range of 0–50 kPa to quantitatively analyze its sensing performance. The corresponding sensitivity (GF) of the PAM@Loofah was obtained by fitting the curve in Figure 4b.The corresponding equation is shown in the following:$$\mathrm{GF}=\frac{{\Delta R/R}_{0}}{P}$$
where R_0_ is the original resistance, and R is the change in resistance. P is the pressure applied to the PAM@Loofah in kPa. In the pressure range of 0–10 kPa, the PAM@Loofah was fitted linearly to obtain a sensitivity of 116.3 Pa^−1^ with a degree of fit of 0.9695. As the pressure increased to 50 kPa, the sensitivity of the PAM@Loofah sensor decreased to 2.6 Pa^−1^ with a degree of fit of 0.9851. This result shows that a robust and highly sensitive 3D conductive network was constructed in the range of 0–50 kPa. As depicted in Figure 4c, the periodic sensing stability was validated at different frequencies (0.40, 0.79, and 1.18 Hz), indicating minimal impact on this value from the compression frequency. Figure 4d illustrates the relative changes in resistance at 2, 5, and 10 kPa, with all at the same frequency. The relative resistance change curves show that the electrical signals of the PAM@Loofah remain consistent for the same pressure. As the pressure continues to increase, the relative resistance decreases. A good response time is also a key performance metric for practical applications as a sensor. As shown in Figure 4e, the response time of the PAM@Loofah sensor was 58 ms, and the recovery time was 68 ms. This result shows that the PAM@Loofah had excellent response time and could feed back the electrical signal in real time.

Sensors are often confronted with thousands of passes in practical applications without being able to experience signal turbulence. The PAM@Loofah was tested for 1000 successive compressions, as shown in Figure 4f, to ascertain the durability and stability of the device. The magnified views of the front, middle, and back of the test in Figure 4f show that the PAM@Loofah had almost the same relative resistance change. This indicates that the PAM@Loofah had excellent durability and could face thousands of times of signal sensing, underscoring its considerable potential for long-term practical applications.

## 4. Applications for PAM@Loofah Sensors

PAM@Loofah sensors were integrated into wearable devices for human motion detection due to their excellent detection of strain generated by pressure. As shown in Figure 5a, the PAM@Loofah was connected to a Bluetooth module, and a signal monitoring system was formed with a smartphone. The smartphone could receive the changes in the PAM@Loofah sensor in real time. In Figure 5b, the device was positioned at the base of the index finger to capture the click action. After the click–release action was completed, the relative resistance changed slightly and then returned to its initial value. The sensing curve of the finger click exhibited a stable periodic resistance response.

In Figure 5c, the device was positioned at the index finger joint to monitor its movement. As depicted, the sensor adeptly captured the curvature of the knuckles. Figure 5 shows the monitoring of wrist motion with the PAM@Loofah sensor, demonstrating that the sensor could accurately monitor and respond to minor wrist flexion and rotation. This result highlights the fast and accurate response of the PAM@Loofah pressure sensor. Furthermore, Figure 5e depicts the electrical signal when the sensor was applied to the palm. When pressure was applied by holding the sensor in the palm of the hand, the electrical signal changed accordingly. The above results show that the sensor had the ability to monitor complex activities.

In Figure 5f, the device was positioned on an elbow of a human body. This allowed for the detection of changes in the electrical signal as the elbow was flexed and extended. Specifically, the resistance decreased when the elbow was flexed and increased when the elbow was extended and stabilized. Figure 5g illustrates the sensor’s installation at the shoulder joint to monitor joint movement. When the sensor was positioned at the junction of the shoulder and the arm, it could capture the periodic clockwise rotation of the shoulder joint as the arm swung downward. The periodic changes in the electrical signals demonstrated the efficacy of the sensors in monitoring the movement of human joints. 

Five PAM@Loofah sensors were connected to a multi-channel receiver and a Bluetooth module, as shown in Figure 6a, to validate the accurate sensing capability of the PAM@Loofah sensors. The occurrence of a gesture was determined by monitoring the relative resistance change of each sensor. Figure 6b shows that all five curves changed simultaneously when a fist was made, and thus, it could be determined that five fingers were bent. Furthermore, when the hand made the OK gesture, only the curves of the thumb and index finger changed. Moreover, when only the index finger was extended, four fingers also underwent bending, showing changes in the corresponding curves. The above results show that the sensing system was able to accurately recognize different gestures with the changes in the curves of the five channels. This holds great promise for applications such as human–machine gesture control, human movement, and health monitoring.

## 5. Conclusions

In conclusion, a feasible method was successfully demonstrated to prepare multifunctional electronic devices using MXene and Ag NWs. The coating of loofah fibers with PDAC facilitated the close adsorption of MXene nanosheets onto their surface through electrostatic forces. When combined with Ag NWs, this process resulted in the formation of a 3D conductive network structure with outstanding conductivity. The synergistic integration of Ag NWs and MXene not only ensured high sensitivity but also provided a wide induction range without compromising mechanical properties. Overall, our study highlighted the potential of MXene and Ag NWs in developing versatile electronic devices with enhanced performance and functionality. During the pressing process, internal MXene nanosheets may crack; however, the external combination of Ag NWs and PDAC phases could maintain connectivity across the cracks, aided by the multitude of internal paths within the loofah. This ensured the integrity of the conductive pathway. The PAM@Loofah sensor fabricated using this method exhibited outstanding pressure-sensing capabilities and could effectively monitor various human movements, including joint bending, facial muscle changes, and subtle pulses. Moreover, the sensor demonstrated remarkable durability, maintaining high sensitivity even after prolonged fatigue testing (1000 cycles). This approach offers a simple and efficient strategy for designing and developing portable multifunctional wearable sensors, presenting novel avenues for simultaneously enhancing the sensitivity and sensing range by selecting sensing materials and conductive networks.

## Figures and Tables

**Figure 1 polymers-16-01377-f001:**
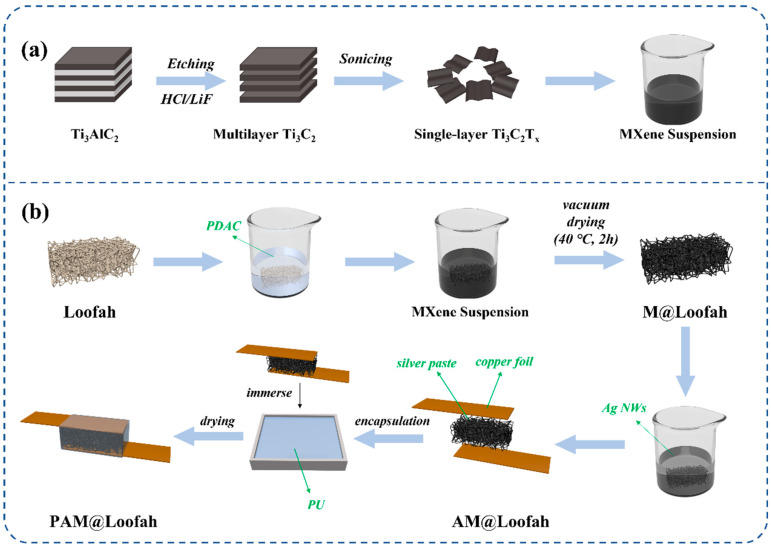
(**a**) Preparation process of single-layer MXene. (**b**) The PAM@Loofah sensor’s assembly process.

**Figure 2 polymers-16-01377-f002:**
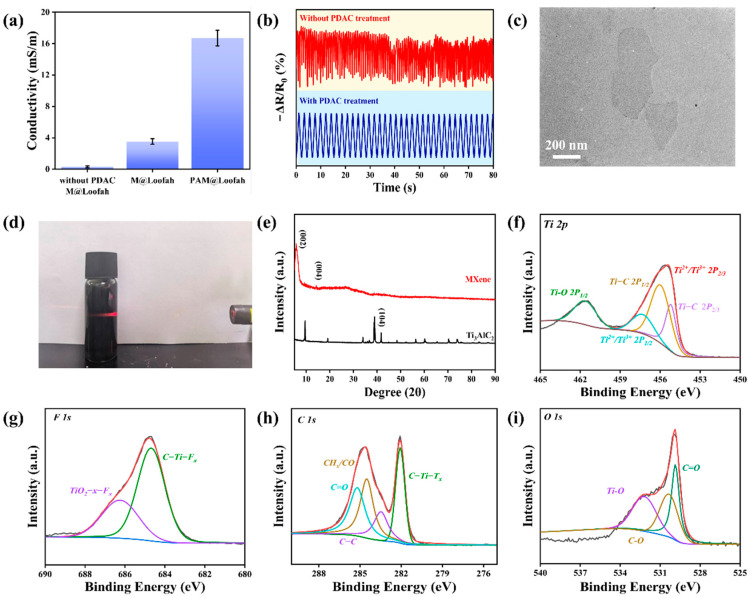
(**a**) Conductivity of the PAM@Loofah and M@Loofah. (**b**) Continuous electrical signal stability testing of sensors with or without PDAC processing. (**c**) TEM images of monolayer MXene nanosheets. (**d**) Pictures of the Tyndall effect in MXene suspensions. (**e**) The XRD patterns of Ti_3_AlC_2_ (MAX). Deconvoluted XPS spectra of (**f**) Ti 2p, (**g**) F 1s, (**h**) C 1s, and (**i**) O 1s of MXene.

**Figure 3 polymers-16-01377-f003:**
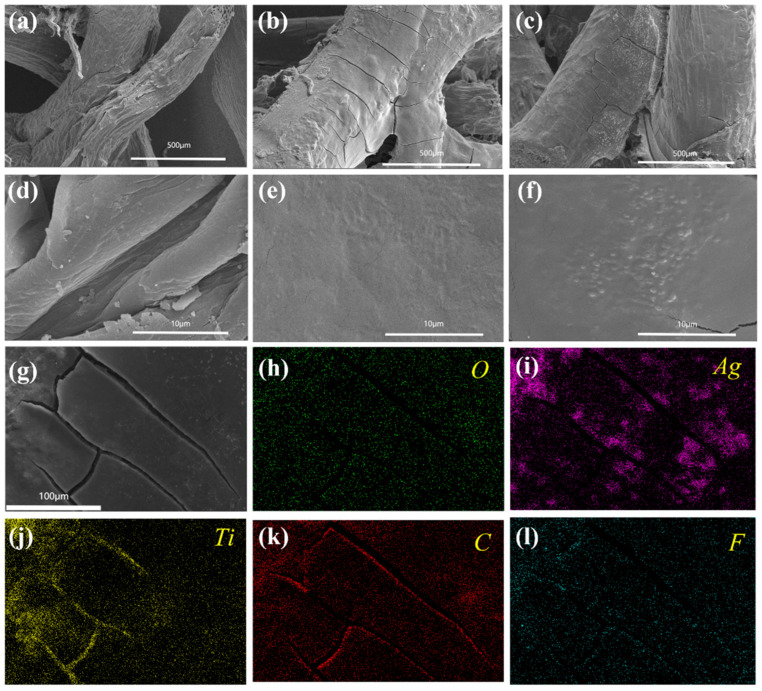
SEM images of (**a**) the pure loofah, (**b**) M@Loofah, and (**c**) AM@Loofah, and (**d**–**f**) show the corresponding high-resolution images, respectively. (**g**) SEM images of the PAM@Loofah and corresponding elemental mapping images of (**h**) oxygen, (**i**) silver, (**j**) titanium, (**k**) carbon, and (**l**) fluorine.

**Figure 4 polymers-16-01377-f004:**
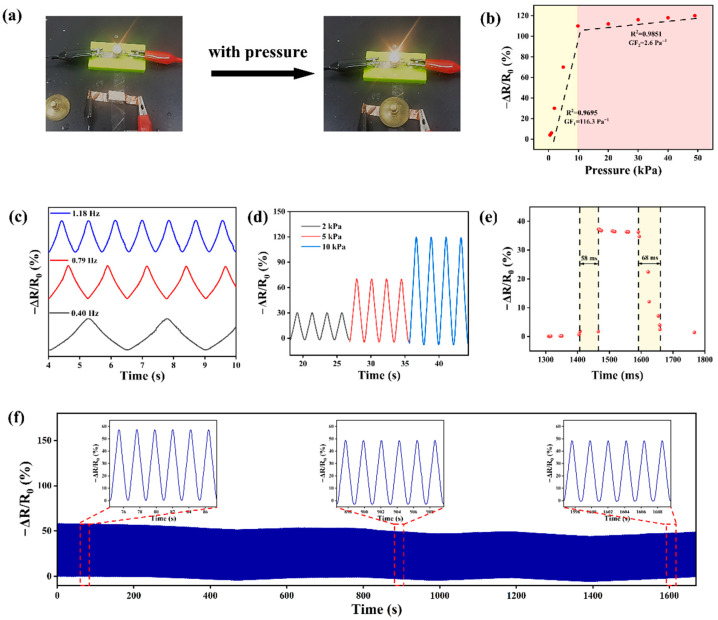
Sensing performance of the PAM@Loofah. (**a**) A picture of a light bulb brightening when the PAM@Loofah sensor was pressurized. (**b**) Relative resistance change in the PAM@Loofah in the pressure range of 0–50 kPa (corresponding sensitivities and fits are shown in the figure). (**c**) Relative resistance change curves of the PAM@Loofah at frequencies of 0.40, 0.79, and 1.18 Hz. (**d**) Relative resistance changes in the PAM@Loofah at pressures of 2 kPa, 5 kPa, and 10 kPa. (**e**) The response and recovery time of the PAM@Loofah sensor under an applied pressure of 3 kPa. (**f**) The stability of the pressure sensor over 1000 cycles at 10 kPa; the inset shows resistance changes over different periods.

**Figure 5 polymers-16-01377-f005:**
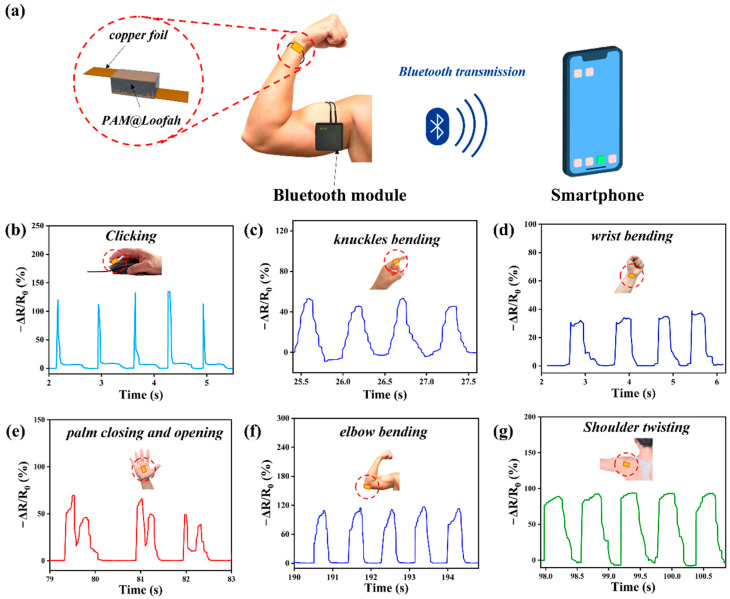
Applications for PAM@Loofah sensors. (**a**) The signal monitoring system involved connecting a PAM@Loofah sensor, a Bluetooth module, and a smartphone. Relative resistance change curves when (**b**) clicking, (**c**) bending the knuckle, (**d**) bending the wrist, (**e**) opening and closing the palm, (**f**) bending the elbow, and (**g**) twisting the shoulder.

**Figure 6 polymers-16-01377-f006:**
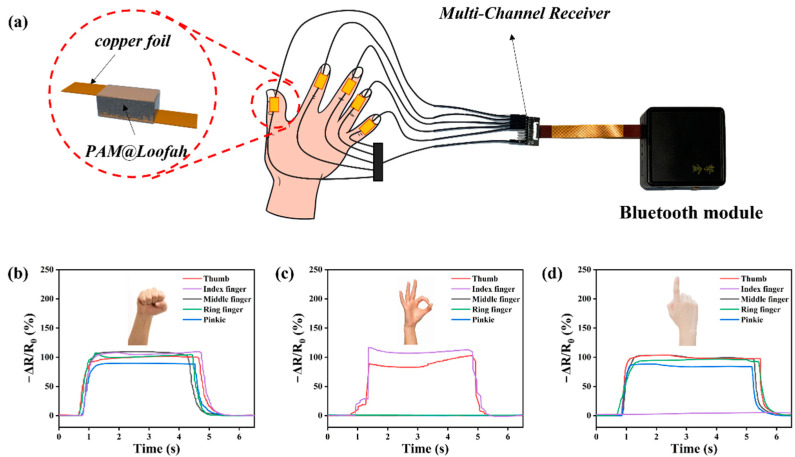
(**a**) The schematic diagram of the assembly of the PAM@Loofah sensor with the multi-channel receiver and the Bluetooth module. Curves of the relative resistance change when the hand made a (**b**) fist and showed the (**c**) OK gesture and (**d**) the index finger.

## Data Availability

The original contributions presented in this study are included in the article, and further inquiries can be directed to the corresponding authors.

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
