# Peer review of "Polyurethane-Encapsulated Biomass Films Based on MXene@Loofah Sponge for Piezoresistive Pressure Sensor Applications"

_polymers, 2024, doi:10.3390/polym16101377_

Round 1

Reviewer 1 Report

Comments and Suggestions for Authors

Comments

The manuscript entitled, ‘Biomass-derived polyurethane 3D films based on Mxene@loofah sponge for piezoresistive pressure sensor applications’ deals about the synthesis of PAM@Loofah for sensor applications. The work is designed well and the characterizations are discussed in details. There are few comments that are given below,

1.       The title says that ‘Biomass-derived polyurethane….’. However, there are no details about the polyurethane either in the materials or experimental section.

2.       The use of the term ‘3D films’ is not suitable, as film represent a 2D structure. Better to modify the title.

3.       In the preparation methods for PAM@Loofah, the concentrations of PDAC solution, MXene suspension, Ag nanowire dispersion should be mentioned. Additional details regarding the final step, i.e., dry curing with polyurethane should be mentioned for clear understanding.

4.       EDS data for PAM@Loofah should be provided to determine the percentage of each elements.

5.       As there could not be any visible differences found in Top-angle SEM images, it will be better to include the cross-sectional SEM images for PAM@Loofah, so that the incorporation of different compounds could be clearly visible.

6.       Characterization details for conductivity measurement is missing.

7.       XPS survey spectrum of MXene should also be included in addition to deconvoluted spectrum.

Author Response

  1. The title says that ‘Biomass-derived polyurethane….’. However, there are no details about the polyurethane either in the materials or experimental section.

Response:

We do not clearly indicate the source of the polyurethane used in our article and apologize for the confusion. The PU (including cross-linking agent) used in this article is not synthesized by ourselves, but purchased from Shenzhen Jipeng Co., Ltd. We have made changes to the title and manuscript content.

Revisions in the manuscript:

Polyurethane-encapsulated biomass films based on Mxene@loofah sponge for piezoresistive pressure sensor applications

Polyurethane (PU) (including cross-linking agent) was obtained from Shenzhen Jipeng Co., Ltd., and model is MR-913 (40% solids).

  1. The use of the term ‘3D films’ is not suitable, as film represent a 2D structure. Better to modify the title.

Response:

We very much apologize for the poor consideration of article topics. Your suggestion makes a lot of sense and we have changed the title of the article accordingly.

Revisions in the manuscript:

Polyurethane-encapsulated biomass films based on MXene@loofah sponge for piezoresistive pressure sensor applications.

  1. In the preparation methods for PAM@Loofah, the concentrations of PDAC solution, MXene suspension, Ag nanowire dispersion should be mentioned. Additional details regarding the final step, i.e., dry curing with polyurethane should be mentioned for clear understanding.

Response:

We sincerely apologize for troubling your identification work by not mentioning the corresponding concentrations in our manuscript. We used 40 wt% aqueous solution of PDAC, 10 mg/mL of MXene suspension, and 1 mg/mL of silver nanowire dispersion. We have modified the article accordingly.

Revisions in the manuscript:

The washed loofah was immersed in PDAC solution (40 wt% aqueous solution) for 2 h and then dried at 40°C. The process was repeated three times. After that, PDAC-treated loofahs were completely immersed in MXene suspension (10 mg/mL) for 10 minutes and then dried in a vacuum oven at 40°C for 2 hours.

Following this, the M@Loofah were completely immersed in an ultrasonic Ag nanowire dispersion (1 mg/mL) for 20 minutes, then removed and dried at 40°C for 10 minutes, re-peating the process twice to get AM@Loofah.

  1. EDS data for PAM@Loofah should be provided to determine the percentage of each elements.

Response:

Thank you very much for your valuable suggestions. We have added the EDS data for each elemental content of PAM in Fig. S3 and Table S1 of the support information.

Revisions in the manuscript:

Fig. S3 shows the EDS image of PAM@loofah demonstrating the content of each element in PAM@loofah. Table S1 shows the specific content of each element in the PAM@Loofah, where Ti and Ag reached 16 wt% and 15.56 wt%, respectively. This result indicates that MXene and Ag were successfully encapsulated onto the loofah. Further, it can be seen in Figs. 3h-l by the distribution of EDS mapping of C,Ti, Ag, F and O that Ag and Ti are even-ly distributed on the loofa. These results demonstrate that MXene and Ag NWs are uni-formly encapsulated on the loofa.

Fig. S3 EDS image of the content of each element in PAM@loofah

Table. S1 Content of each element in PAM@Loofah

Elements

Wt%

Atomic percentage (%)

C

38.37

57.67

N

8.36

10.77

O

12.9

14.56

F

8.81

8.37

Ti

16

6.03

Ag

15.56

2.6

  1. As there could not be any visible differences found in Top-angle SEM images, it will be better to include the cross-sectional SEM images for PAM@Loofah, so that the incorporation of different compounds could be clearly visible.

Response:

Thank you very much for your valuable suggestions. We have retaken the SEM images and presented them in Fig. S2 of the Supplementary Material.

Revisions in the manuscript:

In addition, the SEM image of the PAM@Loofah cross-section in Fig. S2 shows that the conductive network is also formed in the interior of the loofah through the slits.

Fig. S2 SEM image of PAM@Loofah (a) cross-section and (b, c) corresponding high-resolution images.

  1. Characterization details for conductivity measurement is missing.

Response:

We very much apologize for not mentioning the details of the conductivity test. The conductivity of the sensor was calculated based on its resistance and the corresponding dimensional data with the following formulae:

conductivity=GL/A

where G is the conductance of the sensor, and L, A are the length and cross-sectional area of the sensor respectively. We have made corresponding changes in the manuscript to provide details of the conductivity tests.

Revisions in the manuscript:

The conductivity of the sensor was calculated based on its resistance and the corresponding dimensional data with the following formulae:

conductivity=GL/A

where G is the conductance of the sensor, and L, A are the length and cross-sectional area of the sensor respectively.

  1. XPS survey spectrum of MXene should also be included in addition to deconvoluted spectrum.

Response:

Thank you very much for mentioning this question. We have added the X-ray photoelectron spectroscopy (XPS) spectrum of MXene to the support information for the reader's convenience.

Revisions in the manuscript:

Fig .S1 X-ray photoelectron spectroscopy (XPS) spectrum of MXene

Reviewer 2 Report

Comments and Suggestions for Authors

The authors present an interesting study involving the development of sensors based on the 3D structure of a loofah.

The study is very clear overall. The only point that raised some questions involves the understanding of the actual contribution of the MXenes x Ag nanowires on the performance of the sensor. Have the authors tested a sensor only with Ag NW, without the MXene layer? How would it behave, comparatively?

Author Response

Response:

Thank you very much for mentioning this question about the sensor only with Ag NWs. We have tested sensor conductivity with only silver nanowires and without MXene. Due to the low concentration of Ag NW dispersion (1 mg/mL) used, fewer silver nanowires were attached to the prepared sensors. Therefore, the sensor was unstable during signal testing and was not tested further.

Reviewer 3 Report

Comments and Suggestions for Authors

In the proposed manuscript, a feasible method was demonstrated to prepare multifunctional electronic devices using MXene and Ag NWs. Nevertheless, the content of the article and the correspondence of its content to the profile of polymer science raises doubts.

Comments:

1. Both in the title of the article and for the aim of the work, it was indicated that the Ag NWs/MXene@loofah was packaged by polyurethane to form a piezoresistive pressure sensor. However in the experimental part, the packaged of Ag NWs/MXene@loofah by polyurethane is briefly described as “Subsequently, the final pressure transducer performance test was then perforated by dry curing with polyurethane at 40°C for 12 hours.”. In this regard, it should be noted that in the experimental part there is no brand and composition of the polyurethane material used. It is not clear what the dry curing with polyurethane means. It is necessary to show how and using what specific type of polyurethane the treatment was carried out and dry curing with polyurethane. What is the thickness of polyurethane coatings on the surface of the piezoresistive pressure sensor?

2. The introductory part says “Polyurethane (PU) is widely used as a sensor because of its outstanding performance advantages, such as elasticity, flexibility, and durability. [32, 33]." It should be noted here that polyurethanes are used only as binders and not as sensors. There is no mention in references [32, 33] that polyurethanes can be used as sensors on their own.

3. In the introduction there is no mention of the nature, structure and source of purchase of the used loofah. Perhaps for the authors the used loofah seems to be a well-known product, but for a wide range of readers it may be difficult to understand.

4. M@Loofah appeared in Figure 2, and AM@loofah was added in Figure 3. In the experimental part and in the text, no information is given about M@Loofah and AM@loofah. Further clarification is required.

5. The article was submitted to the journal "Polymers". However, the concept of polymers is present only in the title of the article “Biomass-derived polyurethane 3D films based on Mxene@loofah sponge for piezoresistive pressure sensor application.” In what follows, the article discusses only the manufacture of objects of inorganic nature and their properties. The characteristics and any influence of polyurethanes as a polymer binder are not discussed in the article. In this regard, from my point of view, the content of the article is not suitable for publication in the journal “Polymers”.

Author Response

  1. Both in the title of the article and for the aim of the work, it was indicated that the Ag NWs/MXene@loofah was packaged by polyurethane to form a piezoresistive pressure sensor. However in the experimental part, the packaged of Ag NWs/MXene@loofah by polyurethane is briefly described as “Subsequently, the final pressure transducer performance test was then perforated by dry curing with polyurethane at 40°C for 12 hours.”. In this regard, it should be noted that in the experimental part there is no brand and composition of the polyurethane material used. It is not clear what the dry curing with polyurethane means. It is necessary to show how and using what specific type of polyurethane the treatment was carried out and dry curing with polyurethane. What is the thickness of polyurethane coatings on the surface of the piezoresistive pressure sensor?

Response:

We really appreciate you to mention a such meaningful question for us. Also, we apologize for not making it clear in the manuscript that a polyurethane model was used. Firstly, MXene tends to oxidize when exposed to air for a long period of time, resulting in a decrease in the conductivity of the sensor, which seriously affects the signal monitoring ability of sensor. Therefore, purchased polyurethane was used in this manuscript to encapsulate the sensor, thereby insulating it from the effects of oxygen. The polyurethane used in this manuscript is type MR-913 polyurethane (40% solids), purchased from Shenzhen Jipeng Co., Ltd. The thickness of the polyurethane used to encapsulate the sensor was 1-2 mm. We have made corrections to the corresponding places in the manuscript.

Revisions in the manuscript:

Polyurethane (PU) (including cross-linking agent) was obtained from Shenzhen Jipeng Co., Ltd., and model is MR-913 (40% solids).

  1. The introductory part says “Polyurethane (PU) is widely used as a sensor because of its outstanding performance advantages, such as elasticity, flexibility, and durability. [32, 33]." It should be noted here that polyurethanes are used only as binders and not as sensors. There is no mention in references [32, 33] that polyurethanes can be used as sensors on their own.

Response:

Thank you very much for mentioning this valuable question. Polyurethane itself cannot be used as a sensor, but it is characterized by high elasticity, high abrasion resistance, and high impact resistance. In addition, the curing process of polyurethane is very convenient, making it suitable for sensor encapsulation. We apologize for any inconvenience caused by our expression. We have made the appropriate changes in the text to avoid ambiguity.

Revisions in the manuscript:

Polyurethane (PU) is widely used as a sensor package as well as a substrate because of its outstanding performance advantages, such as elasticity, flexibility, and durability.

  1. In the introduction there is no mention of the nature, structure and source of purchase of the used loofah. Perhaps for the authors the used loofah seems to be a well-known product, but for a wide range of readers it may be difficult to understand.

Response:

We apologize for the confusion caused to our readers by not mentioning where to purchase loofahs in the experimental section. To improve the ease of preparation of this sensor, the loofah sponges we used were from commercially available unmodified natural loofah products, including but not limited to cleaning tools such as brushes. We have made corresponding changes in the manuscript to provide purchasing information for the loofah sponges used.

Revisions in the manuscript:

Natural loofah sponge bought from YANCHEG OUKAI sponge products factory.

  1. M@Loofah appeared in Figure 2, and AM@loofah was added in Figure 3. In the experimental part and in the text, no information is given about M@Loofah and AM@loofah. Further clarification is required.

Response:

We apologize for any confusion caused to our readers by not expressing the preparation process of M@Loofah and AM@Loofah clearly. M@Loofah is a loofah sensor treated only with MXene immersion. AM@loofah is a loofah sensor treated with MXene and Ag NWs. PAM@loofah is an AM@loofah sensor after encapsulation with polyurethane. We have modified the experimental section accordingly.

Revisions in the manuscript:

The washed loofahs were soaked in a solution of PDAC (40 wt% aqueous solution) and dried at 40°C for three cycles. Next, PDAC-treated loofahs were completely immersed in MXene suspension (10 mg/mL) for 10 minutes and then dried in a vacuum oven at 40°C for 2 hours. Repeat this process twice to get M@Loofah. Following this, the M@Loofah were completely immersed in an ultrasonic silver nanowire dispersion (1 mg/mL) for 20 minutes, then removed and dried at 40°C for 10 minutes, repeating the process twice to get AM@Loofah.

Subsequently, the AM@Loofah was immersed in polyurethane with curing agent and cured at 40°C for 12 hours. Finally, the cured polyurethane was cut and trimmed to obtain PAM@Loofah.

  1. The article was submitted to the journal "Polymers". However, the concept of polymers is present only in the title of the article “Biomass-derived polyurethane 3D films based on Mxene@loofah sponge for piezoresistive pressure sensor application.” In what follows, the article discusses only the manufacture of objects of inorganic nature and their properties. The characteristics and any influence of polyurethanes as a polymer binder are not discussed in the article. In this regard, from my point of view, the content of the article is not suitable for publication in the journal “Polymers”.

Response:

Thank you very much for mentioning this valuable question. Based on our understanding, we have put forward the following opinions:

First, according to the submission scope of Polymers, "Applications of Polymers: all kinds of applications (from sensors to actuators, from biomedical engineering to space engineering, and from the macro scale down to the nano scale) with polymeric materials, proof of concept, structural/system design, performance verification and characterization." In this manuscript, we used polyurethane encapsulated sensors, which play a crucial role in the preparation process. MXene is prone to oxidation when exposed to air for a long time. This can cause a decrease in the conductivity of the sensor and a disturbance in the sensing signal. After encapsulating the sensor with polyurethane, it can ensure the long-term stable use of the sensor. Therefore, we believe that this belongs to the application of polymers in sensors.

Secondly, in the scope of submission, it was also mentioned that "Polymer Processing and Engineering: shaping, synthesis, transformation, compounding, functionalization, and stabilization of polymeric materials, including extrusion, film blowing, wire-coating, injection molding, blow molding, thermoforming, calendaring, mixing, compression and transfer molding, rotational molding, plastic foam molding, reactive polymer processing, fiber spinning, additive manufacturing and 3D printing."

Polyurethane is widely used as a substrate material in the preparation process of sensors. In addition, the mechanical properties, response time and signal stability of the sensors tested in the manuscript are directly related to the properties of polyurethane. The sensor was developed by combining the advantages of polymers and inorganic materials. Therefore, we believe that this manuscript belongs to the application of polymer material sensors and includes molding and composite with inorganic materials.

Round 2

Reviewer 1 Report

Comments and Suggestions for Authors

All the given comments are answered and incorporated in the manuscript. The manuscript can be accepted for publication in the present form.

Reviewer 3 Report

Comments and Suggestions for Authors

The reviewer's comments have been taken into account, there are no further comments.